# Self-Limitations of Heat Release in Coupled Core-Shell Spinel Ferrite Nanoparticles: Frequency, Time, and Temperature Dependencies

**DOI:** 10.3390/nano11112848

**Published:** 2021-10-26

**Authors:** Shankar Khanal, Marco Sanna Angotzi, Valentina Mameli, Miroslav Veverka, Huolin L. Xin, Carla Cannas, Jana Vejpravová

**Affiliations:** 1Department of Condensed Matter Physics, Faculty of Mathematics and Physics, Charles University, Ke Karlovu 3, 121 16 Prague 2, Czech Republic; skhanal@mag.mff.cuni.cz (S.K.); miroslavveverkag@gmail.com (M.V.); 2Department of Chemical and Geological Sciences, University of Cagliari, S.S. 554 bivio per Sestu, 09042 Monserrato, CA, Italy; sannamarco91@gmail.com (M.S.A.); valentina.mameli@unica.it (V.M.); 3Consorzio Interuniversitario Nazionale per la Scienza e Tecnologia dei Materiali (INSTM), Via Giuseppe Giusti 9, 50121 Firenze, FI, Italy; 4Department of Physics and Astronomy, University of California, Irvine, CA 92617, USA; huolinx@uci.edu

**Keywords:** core-shell nanoparticles, magnetic fluid hyperthermia, frequency dependence, time dependence, temperature-dependence, blocking temperature

## Abstract

We explored a series of highly uniform magnetic nanoparticles (MNPs) with a core-shell nanoarchitecture prepared by an efficient solvothermal approach. In our study, we focused on the water dispersion of MNPs based on two different CoFe_2_O_4_ core sizes and the chemical nature of the shell (MnFe_2_O_4_ and spinel iron oxide). We performed an uncommon systematic investigation of the time and temperature evolution of the adiabatic heat release at different frequencies of the alternating magnetic field (AMF). Our systematic study elucidates the nontrivial variations in the heating efficiency of core-shell MNPs concerning their structural, magnetic, and morphological properties. In addition, we identified anomalies in the temperature and frequency dependencies of the specific power absorption (SPA). We conclude that after the initial heating phase, the heat release is governed by the competition of the Brown and Néel mechanism. In addition, we demonstrated that a rational parameter sufficiently mirroring the heating ability is the mean magnetic moment per MNP. Our study, thus, paves the road to fine control of the AMF-induced heating by MNPs with fine-tuned structural, chemical, and magnetic parameters. Importantly, we claim that the nontrivial variations of the SPA with the temperature must be considered, e.g., in the emerging concept of MF-assisted catalysis, where the temperature profile influences the undergoing chemical reactions.

## 1. Introduction

Magnetic nanoparticles (MNPs) own magnificent properties exploitable in various applications, such as magnetic fluid hyperthermia (MFH), magnetic resonance imaging (MRI), targeted drug delivery, material science, and much more [1,2,3]. In addition, targeting and regulating the movement of MNPs with external agents, such as magnetic field, not only enable the suppression of tumor cells locally, but also helps to repair the damaged tissues without affecting neighboring healthy cells [4,5,6]. In the presence of external high-frequency AMF (in the kHz range), heat generated from the MNPs effectively destroys deep-rooted tumor cells [7,8].

Magnetic heating has recently been demonstrated as an efficient way to perform catalytic reactions as the MNPs can be used for heating the reaction mixture [9,10]. After deposition of the heating agent and the catalyst on a support, the AMF heating enables to carry out transformations that are otherwise performed heterogeneously at high pressure and/or high-temperature conditions [11,12]. The reason for this unique reactivity is the fast heating of the MNPs well above the boiling temperature of the solvent and the local creation of “hot spots”, surrounded by a vapor layer–nanoscopic reaction vessels, in which high temperature and pressure may be present. Considering the nature of the process, precise control of the heating profile is necessary, which can be achieved by the smart configuration of the used heating agents.

In high-frequency AMF, there are three main mechanisms to release heat energy from MNPs: hysteresis losses, superparamagnetic (SPM) relaxation, and friction [13,14,15].

In the SPM regime, the particle’s macrospin reversal occurs via Brownian and/or Néel relaxation mechanisms represented by the Brownian (τ_b_) and Néel (τ_n_) relaxation time (for details, please, see Appendix A) [7], and they both contribute to the frequency- and amplitude-dependent heat dissipation [16,17,18]. The most straightforward approaches to describe AMF-induced heating are based on Stoner–Wohlfarth and Linear Response Theory (LRT) [19,20]. Nevertheless, they neglect the motion of the MNPs in the fluid. More sophisticated models based on stochastic theories incorporate both mechanisms, giving analytical solutions only for some limit cases [21,22,23]. For moderate AMF, the Néel relaxation is not disturbed, and the LRT employs the effective (resulting) relaxation based on the Brownian and Néel process, assuming to be independent [21,23]. Consequently, MNPs with high values of *K* feature significant Brownian contribution, while the moderate-*K* MNPs reveal the Néel mechanism as dominant.

Within the LRT, the heating abilites of the MNPs can be expressed by the power loss density, *P* is related to the specific power absorption, SPA as SPA= *P*/*ρ*, where *ρ* is the mass density of the particles. Note that the SPA is also known as specific loss power, *SLP* or specific absorption rate, *SAR*. Consequently, *P* and, thus, the SPA is linearly related to the particle volume and quadratically with the amplitude of the applied field.

It should be noted that the dominant relaxation mechanism (represented by an effective relaxation time, *τ*_r_ given by Equation (S11)) is mainly due to the “faster mechanism”. As discussed above, the *τ*_b_ is mostly dominating for systems with large anisotropy and low viscous solvents and vice versa for the *τ*_n_. It is also worth mentioning that Néel and Brown relaxation contributions to *τ*_r_ considerably depend on the particle size in addition to the effective anisotropy constant.

A more complicated equation for the SPA, taking into account the particle size distribution, was introduced by Rosenweig [13] and adopted, e.g., by Torres and coworkers [24]:(1)$$\mathrm{SPA}\left(\langle 2r\rangle \right)=\int_{0}^{\infty }\mathrm{SPA}\left(2r\right)g_{\sigma }\left(2r\right)dr=\frac{1}{\varrho }H^{2}f\pi \mu_{0}\int_{0}^{\infty }\frac{Z}{{\left(1+Z\right)}^{2}}\chi_{0}g_{\sigma }\left(2r\right)dr,$$
where *ρ* are the MNPS’ density and *g*σ (2*r*) is the distribution function representing the particle size distribution. For example, Torres and co-workers used Gaussian distribution [24], while log-normal distribution is expected for a typical ensemble of MNPs [19,25].

The parameter *χ*_0_ in Equation (1) represents the equilibrium susceptibility of the SPM sample [13]. As an approximation, it can be considered as the static volume susceptibility per MNP given by the well-known Langevin function.

In addition, the material’s parameters can be embedded in the prediction of the heat release. For a given ensemble of MNPs, there is a certain threshold of frequency for a given *H*, where the MNPs can achieve the maximum heat release. The corresponding SPA vs. *f* dependence can be calculated using the formula:(2)$$\mathrm{SPA}=\Gamma H_{0}^{2}\frac{2\pi \tau_{r}f^{2}}{{\left(2\pi \tau_{r}f\right)}^{2}+1},$$
where Γ is a field- and frequency-independent parameter that is given by the material’s properties of the MNPs, i.e., critical monodomain’s size, which depends on the intrinsic magnetic anisotropy and stiffness [26]. Equation (2) suggests that there is always an upper bound for the SPA determined by the parameter Γ, which represents the unique MNP’s properties.

With respect to the rational material’s design, one of the most significant assets of MNPs is that their properties can be tailored on demand. Shape, size, size distribution, and chemical nature can be easily tuned artificially to modulate static and dynamic magnetic responses, which provides extra freedom to use them in a wide range of applications [27]. In that regard, numerous studies explore the heating response of various types of MNPs and try to correlate their material’s parameters to the observed heating performance, e.g., [2,28,29]. Among them, magnetite and maghemite are the most studied architectures because of their biocompatibility and convenient magnetic parameters [6,30]. By replacing the iron cations with other ions such as Co^2^^+^, Mn^2+^ and Ni^2+^ open up further potential to tune the saturation magnetization (*M*s), magnetic anisotropy constant (*K*), coercivity (*H*_c_), blocking temperature (*T*_b_) and, thus, ultimately enhance the heating response [31]. Nevertheless, tuning the magnetic parameters via chemical substitution and particle size is limited by the bulk value of *K* and MNP’s volume.

Additional variability of the MNPs’ design can be introduced by combining two (or more) different spinel ferrite phases in a single MNP, e.g., by covering a spinel ferrite MNP by an extra shell formed of a different spinel ferrite. Such core-shell MNPs offer a great potential for fine-tuning all the material’s parameters to reach outstanding heating properties, which can be optimized using hard (h) and soft (s) spinel ferrite phases [32,33,34].

Although accurate control over the MNPs’ properties is possible, the resulting heat release originates from a delicate balance between the high-frequency MF parameters, viscosity, and heat capacity of the liquid medium, as well as the chemical, structural, morphological, and magnetic properties of the used MNPs discussed above [35]. On top of that, the mesoscopic effects, such as the formation of chains or aggregation, which can be reversible or irreversible, significantly impact the heating efficiency [28,36]. It has also been reported that the enhanced heating by magnetically well-oriented samples is due to the greater effective anisotropy energy density [15,28] along with the linear chain-like structures, which is the origin of the larger dynamic hysteresis loop area [37]. Moreover, the mesoscopic changes in the MNP’s architecture within the dispersions are closely related to the concentration [38] and surface coating [39,40], giving rise to the competition of steric and electrostatic effects modifying the stability of the dispersion. Finally, the experimental values are strongly influenced by the error of temperature read-out, aging effects, and deviation from the adiabatic conditions. Therefore, absolute values measured on different experimental setups on exactly the same dispersion may vary significantly. In spite of all these internal and external hitches, most strategies still base on the straightforward correlation of the MNPs’ size and the heating performance [41,42].

Experimentally, the most common practice to evaluate heating efficiency of MNPs, to date, is by considering the initial heating phase for the calculation of the SPA and Intrinsic Loss Power (ILP) [30]. Even though initial SPA is widely adopted to signal the heating efficiency of MNPs, it does not fully represent their heating efficiency in the whole temperature range of action, especially for the MNPs whose magnetic properties are sensitive to temperature and magnetic state phase transition occurring during the heating process (such as switching from the blocked state to the SPM state).

For example, the so-called self-control hyperthermia profits from this mechanism, and the material is chosen to have the Curie temperature in the coveted range to suppress the heat release due to ferromagnetic to paramagnetic phase transition [43]. Nevertheless, the non-linear behavior of the heating curves of many common and core-shell MNPs suggests that monitoring the time evolution of the heat release with respect to the applied frequency and amplitude of the AMF is of utmost importance, although not much investigated.

To address this peculiar problem, the evolution of the SPA with AMF frequency, time, and the temperature has to be studied meticulously. This work focused on water dispersions of highly uniform hard/soft bi-ferrimagnetic core-shell MNPs. The series of MNPs was subjected to comprehensive structural and magnetic characterization in the previous study [44]; an in-depth insight in the heating abilities is the main target of the current study. We present a thorough study of their heating response and address the intrinsic non-linearities during the heating process. Our study features for the first time the importance of time and temperature dependence during the high-frequency field stimulation of MNPs, causing switching between the blocked and SPM regimes and the interplay between the Néel and Brownian mechanism as a function of temperature.

## 2. Materials and Methods

A representative series of hydrophilic core-shell MNPs with narrow CoFe_2_O_4_ core size and shell thickness distributions was prepared using two-step hydrothermal synthesis reported previously [45]. First, the hydrophobic oleate-coated core-shell MNPs were built from two different sizes of the oleate-coated CoFe_2_O_4_ MNPs (termed Co1 and Co2) followed by the growth of spinel ferrite shells: Mn = MnFe_2_O_4_ and Fe = γ-Fe_2_O_3_ (termed Co1@Mn, Co1@Fe, Co2@Mn, and Co2@Fe). The hydrophobic MNPs were made hydrophilic by an intercalation process with cetyltrimethylammonium bromide (CTAB, (C_16_H_33_)N(CH_3_)_3_Br) [44,46]. The hydrophilic MNPs were dispersed in water and the concentration of MNPs in water dispersions was fixed to 3.4 mg/mL for all samples to avoid the concentration effects in the heating experiments.

The samples were first characterized by high-resolution transmission electron microscopy (HRTEM) using FEI Talos F200X (Thermo Scientific™, Waltham, MA, USA) with Schottky-field emission gun operating at 200 kV to collect HRTEM images. Nanoscale chemical mapping was carried out by STEM-EELS and EDX (JEOL 2100F, Tokyo, Japan). Powder X-ray diffraction (XRD) was carried out on the PANalytical X’Pert PRO (Malvern Panalytical, Malvern, UK) with Cu Kα radiation (1.5418 Å), equipped with a secondary monochromator and a PIXcel position-sensitive detector. The powder XRD data were analyzed using standard Rietveld analysis with the help of FullProf software, yielding the lattice parameter (a) and the mean MNPs’ diameter (D_XRD_). Magnetic measurements of the powders and dispersions were carried out using a SQUID magnetometer (MPMS7XL, Quantum Design, San Diego, CA, USA). All data were corrected according to the organic content obtained by the thermogravimetric analysis (TGA, Mettler Toledo, Columbus, OH, USA). The magnetic parameters, including mean magnetic moment per particle (μ_m_, Equation (S2)) and the corresponding magnetic size (D_MAG_, Equation (S3)), were obtained using the procedures respecting the real nature of the MNPS’ ensembles (assuming log-normal distribution of μ, *T*_b_, etc.); details of the data processing are given in the Appendix A. Hydrodynamic diameters (D_H_) were studied using dynamic light scattering technique (DLS) with the help of ZetaSizer device (Malvern Panalytical, Malvern, UK).

The heating response of MNPs’ water dispersions was recorded using a D5 system (nanoScale Biomagnetics, Zaragoza, Spain) in the frequency range 159–782 kHz and amplitude of 31.6 mT. A fiber optic probe immersed in the dispersion was used to monitor the temperature of the solvent during the experiment. Importantly, the frequency dependence of the heating response was recorded under the adiabatic condition ensured by decoupling the sealed vial with the optical sensor from surroundings by placing in a glass jacket evacuated by a turbomolecular pump. The SPA was evaluated using the well-known formula:(3)$$\mathrm{SPA}=\frac{C\delta }{\phi }\frac{dT}{dt},$$
where *C* and *δ* are specific heat capacity and density of solvent respectively with ϕ as weight concentration of MNPs in the colloidal dispersion, *T* is the actual temperature of the system, and *t* is the time. The heating curves for the highest and lowest applied frequency were reproduced and compared to the first set of measurements to ensure robustness for the results; only the data sets with agreement > 90% were accepted.

## 3. Results and Discussion

### 3.1. Basic Characterization

In our recent work [44], the exchanged-coupled core-shell spinel ferrite MNPs have been investigated, focusing on the effect of the core size, the chemical nature of the shell, and the shell thickness on the initial heating abilities for a single frequency and amplitude of the applied AMF. The samples were fully characterized for their structural, morphological, and magnetic properties. The formation of well-separated NPs due to the presence of oleate molecules as capping agent, and the core-shell architectures were proved by TEM and HR TEM and nanoscale chemical mapping by STEM-EELS and EDX, as shown in Figure 1. The HRTEM images reveal a high uniformity of the prepared core-shell MNPs; the inset in Figure 1a demonstrates that the crystallites are well-developed. In addition, the HRTEM FFT image confirmed the presence of the spinel ferrite phase, and the sharp spots in the pattern corroborate the high crystallinity. After intercalation of the CTAB, no clustering phenomena was observed (please, see Appendix A). The XRD patterns also served to confirm the phase composition and to evaluate the D_XRD_ as a measure of the structural coherence [44]. Finally, static magnetic characterization (zero-field cooled (ZFC) and field cooled (FC) temperature dependence of magnetization and magnetization isotherms [44]) revealed important information about the magnetic properties in the blocked and SPM state.

Selected structural and magnetic parameters are visualized in Figure 2, and the original data are summarized in Appendix A in the Appendix A. It is worth mentioning that all MNPs have quite high and similar values of the *M*s, which points to the overall high crystallinity of MNPs [47].

Figure 2a shows the equilibrium blocking temperatures; the *T*_b_ corresponds to the mean value based on the Equation (S5), while the *T*_b,diff_ is the temperature of the ZFC-FC furcation point. Parameters of the characteristic macrospin’s relaxation are given in Figure 2b. Considering the range of MNPs’ size and effective anisotropy constants, the Néel relaxation is expected to be dominating over Brownian relaxation in our case [44,46]. Nevertheless, considering the hydrodynamic size, D_H_ of the ferrite MNPs in water (Appendix A), we can also estimate the Brownian and effective relaxation times using Equations (S10) and (S11).

An essential parameter is the mean magnetic moment per MNP, presented in Figure 2c. This value is directly related to the magnetically active volume of the MNP, and as it will be shown further, it plays an important role in the heating properties. Finally, the diameters of the MNPs obtained by the different methods are summarized in Figure 2d.

A first inspection of the data suggests that there are rather moderate variations in the *T*_b_ and various MNPs’ diameters, while the τ_n_ and *μ*_m_ show a higher diversity. This observation has some added value for discussing the heating properties of the MNPs, in particular, for identifying the most important material’s parameters influencing the heating response.

### 3.2. Heating Properties

In this section, the core results of our study, heating performance of the MNPs in the AMF, will be discussed. Note that for the extensive evaluation of the frequency and time dependence, we employed samples with two different core sizes and two different types of spinel ferrite coating, dispersed in water at the same concentration.

First, we validated that the parameters of the MNPs and the experimental conditions can be understood within the above-discussed models. We simulated the SPA vs. frequency curves using Equation (2) assuming Γ*H*_0_^2^ = 1. This presumption, however, neglects the influence of the intrinsic material’s parameters, and thus the resulting curves depend only on the characteristic relaxation time, τ_r_. According to this model, we should expect a monotonous evolution of the SPA with frequency and a rather moderate variation of the absolute values (Appendix A).

We further considered the more realistic form of the SPA for a real ensemble of MNPs (Equation (1)), assuming the log-normal distribution of the magnetic moments. We substituted the d_0_ by the experimentally derived magnetic size, D_MAG_ (given in Appendix A)_,_ based on the mean magnetic moment values (given in Appendix A). The results of the simulations are shown in Appendix A. They suggest that we can expect some variations in the SPA dependencies among the samples and the most pronounced evolution with the applied frequency is predicted for the Co2@Fe.

The experimental heating curves for the core-shell samples, collected at frequencies 159, 305, 384, 497, 639, and 782 kHz with AMF field amplitude 31.6 mT, are presented in Figure 3a–d (results for the Co1 and Co2 cores are presented in Appendix A). It is worth of mentioning that the original cores (Co1 and Co2) show smaller ∆T (determined as a difference between the initial temperature and the final temperature after 600 s) comparing to the core-shell architectures. The effect is more pronounced for the Co2 series, where the ∆T recorded at the maximum frequency reaches about 10 K, while the core-shell MNPs reveal about 4–5 times larger values. The observed effect is usually attributed to the exchange coupling of the spins in the shell to the macrospin of the core giving rise to the enhanced heating performance [33].

Inspecting the core-shell MNPs, maximum ∆T is observed for the sample Co2@Fe, in agreement with the previous results obtained at 187 kHz and 21 mT [44]. In addition, minor kinks have been observed in the temperature versus time graphs. Inspecting the experimental heating curves more in detail, the initial phase of the heating reveals more complex behavior with respect to the applied frequency and time elapsed. For instance, at the AMF’s frequency and amplitude of 639 kHz and 31.6 mT, a kink has been observed at around 140 s for Co1@Mn and Co2@Mn (Figure 3b,d). In addition, kinks in the time dependent heating curves are observed for both Co1@Mn and Co2@Mn for different frequencies values (Figure 3a,c). In addition, the sample Co2@Fe show two different regimes. First, there is a rapid increase of the temperature at the beginning of the heating process (comparing to the other samples), and then the heating curves tend to saturate and a sort of plateau is reached for the highest frequency. This behavior is somewhat consistent with the predictions based on Equation (1) (Appendix A). On the contrary, in the heating curves of the other samples a rather constant increase with much less pronounced trend to saturation is observed.

To unfold the nature of temperature ramping with time and to quantify and compare the heating efficiency of MNPs in the presence of the AMF, the SPA of the MNPs has been calculated using Equation (3). First, the SPA vs. time has been evaluated for a constant amplitude of AMF (31.6 mT) at varying frequencies; the resulting dependencies are shown in Appendix A for the cores and core-shell samples, respectively. A typically higher than expected SPA values have been observed at lower frequencies at the initial stage of heating except for the Co2@Fe. From the SPA vs. time dependencies (always considered at the initial heating phase), numerical values of the SPA have been evaluated (data at 305, 497, and 782 kHz are presented in Appendix A).

Let’s focus first on the initial heating phase. For the Co1 series, the Co1@Mn has higher SPA (439 W/g) compared to the Co1@Fe (224 W/g) at frequency and field of AMF 305 kHz and 31.6 mT, respectively. Nonetheless, the Co2 series shows an opposite trend with much closer SPA values. When the frequency of the AMF increased to 497 kHz, the SPA values do not increase for all samples, as expected from the theoretical predictions. For instance, for Co1 series, SPA has actually decreased with the frequency; in contrast, SPA has increased for Co2 series. Upon further increase of the frequency, the SPA increases for all samples except for Co2@Mn. These inconsistent variations of the SPA with frequency suggest that stereotyping the heating capability of MNPs via the initial SPA returns an incomplete picture.

Being aware that magnetic properties and relaxation mechanisms of MNPs also depend on the temperature, the variation of the SPA with temperature has been evaluated; the results are shown in Figure 4 for the core-shell samples (the SPA vs. *T* curves of the cores are shown in Appendix A).

For the Co1@Mn and Co1@Fe samples (Figure 4a,b), this kind of “double slope” behavior exists at the frequencies 159 and 305 kHz and disappears when the frequency increases to 384 kHz. In addition, higher SPA values have been observed at lower frequencies at the initial heating phase at the “double slope” range of frequencies. A similar nontrivial trend of the SPA vs. temperature has been observed for Co2@Mn, shown in Figure 4d. Importantly, the “double slope” behavior occurs at higher frequencies comparing to the Co1 series and vanishes for the highest applied frequency of 782 kHz. On the contrary, Co2@Fe exhibits an almost linear behavior, in agreement with the different behavior of SPA vs. frequency, although the curves at 305 and 639 kHz appear not parallel to the others.

Note that we used water dispersions with the same concentration of MNPs, whose hydrodynamic sizes are not expected to dramatically deviate among the series. In addition, a rational assumption is that the D_H_ does not change significantly within the inspected interval (to verify this statement, we carried out a temperature-dependent DLS measurement presented in Appendix A). Thus, the *T*-dependence of the Brownian relaxation is mostly given by the *T*-variation of the water viscosity, which is clearly monotonous and relatively moderate within the temperatures of our interest. Therefore, the peculiarities of the heat release revealed by the experimental heating curves and the corresponding SPA vs. *T* at different frequencies must be predominantly ascribed to the *T*-dependent interplay of the crucial magnetic parameters (*K*, D_MAG_, etc.) of the MNPs.

### 3.3. Heating Abilities in the Context of MNPs’ Parameters

A final picture of the heating abilities represented by the SPA and ∆*T* at the varying frequency for the core-shell samples are presented in Figure 5 (data for the cores are shown in Appendix A). While the ∆*T* is increasing uneventfully with frequency, the SPA shows clear anomalies. In addition, Co1Mn reveals higher ∆*T* than Co2Mn, which is coherence with the trend obtained for the original cores (Co1 and Co2). As opened by the discussion of the SPA vs. *T* dependencies, to get a realistic insight into the heating response of the MNPs, it is inevitable to explore the magnetic properties, and in particular, those governing the SPM relaxation.

First, of all, the SPM relaxation is temperature (and also frequency) sensitive. For example, recent computational study has shown that the self-heating core and shell of core-shell MNPs may respond differently with the temperature [48].

We have to consider the effect of coupling between the constituting spinel phases (s and h). The type of coupling between the core (*h*-phase) and shell (*s*-phase) depends on the thickness of the shell along with the characteristic thickness of the domain wall of a bulk ferrite phase [49]. Considering these parameters in our samples (∆_TEM_ = 1.4–2.9 nm; domain wall width ~ tens of nm [33]), the rigid coupling occurs between the magnetization of the core and the shell. Therefore, only a single switching field for the whole core-shell nanoarchitecture is expected, and the complex evolution of the SPA with time (temperature) and frequency must have a different origin (Figure 5).

Existence of two maxima on the imaginary part of the a.c. susceptibility (also present on some of our samples, Co2@Mn and Co2@Fe, as demonstrated in our previous study [44]) is an indication of the fact that multiple nonequilibrium transitions occurred during the heating. The absence of double peaks in the other samples, both in powder and dispersion, does not permit to ascribe the double-slope behavior to it, being present in all the samples.

Now, we can inspect the key magnetic parameters in more detail. A very important constraint is the blocking temperature; however, for a real ensemble of MNPs one has to consider its distribution due to the MNPs’ size and collective effects. We addressed this fact by defining a mean value, *T*_b_ and an upper bound value, *T*_b,diff_ (details on the procedure are given in the Appendix A and in [44]); the actual values are given in Appendix A, and the graphical presentation is shown in Figure 1a. Comparing the values for the two series based on the different cores, Co1 and Co2, the less heating series features overall larger *T*_b_ and *T*_b,diff_. A similar trend can be observed for the physical size of the MNPs, D_TEM_, which is widely used as the key parameter of the MNPs for the correlation of the SPA [42]. Nevertheless, the D_XRD_ and D_MAG_ follow the trend for the observed heating abilities matching with the best heating properties of the Co2@Fe sample. This particular observation corroborates the disqualification of the physical size as a relevant parameter for prediction and standardization of the heating performance.

Ruta and coworkers offered an intricate theoretical picture pointing to the interplay of Néel and Brownian relaxation, hysteresis losses, and collective effects, such as dipolar interactions, which are beyond the abilities of the actual theoretical efforts [14,50]. Nevertheless, their models clearly conclude that for a given frequency and amplitude of the AMF and magnetic anisotropy, an optimum MNP’s size reveals the highest SPA, which occurs in a transitional regime between the blocked and SPM states. The maximum heating efficiency will be reached when the time of the dominant relaxation process matches the characteristic time of the hyperthermic measurement, τ_SAR_. Therefore, MNPs with a τ_r_ matching the τ_SAR_ must be more efficient.

Ota and Takemura suggested an empirical law pointing to the dominance of the Néel regime when the difference between τ_n_ and τ_b_ is negligible because of the large anisotropy energy and small random torque caused by thermal disturbances. They also reported that the effect of dipole interactions is illustrated as a reduction in the magnetization in the Néel regime, whereas the magnetization derived from the Brownian regime was not affected by dipole interactions [51]. Although the Néel relaxation is accepted to dominate in the MNPs with the D_MAG_ below ~15 nm [30,52], the Brownian motion may also concur to the heat release. Considering the typical values of D_H_~30 nm, a certain contribution of the Brownian mechanisms is expected. It should also be mentioned that the critical diameter at which τ_n_ = τ_b_ for not-agglomerated single-phase cobalt ferrite MNPs in water is ~7 nm [20].

In addition, the *T*-dependence of the Brownian relaxation time is much less sensitive to the magnetic parameters comparing to the Néel relaxation time. In this vein, we should also inspect the temperature dependence of the magnetic monodomain’s size, represented by the D_MAG_ value and the *K*. As reported by Garaio and coworkers; both parameters show about a 20% decrease in the interval 300–350 K for maghemite MNPs with size 12–16 nm [53]. As the τ_b_ is modified only by the *V*, while the τ_N_ depends both on the *K* and actual “magnetic volume”, the critical magnetic size at the *T*_b_ is worth inspection.

The critical diameters, *d*_c_, were calculated using a phenomenological relation (Equation (S7)), and the results are summarized in Appendix A. The largest and very close values were obtained for the Co2@Fe and Co2@Mn samples. This result, however, is not as trivial as the *T*_b,diff_ is larger for the Co1@Fe and Co1@Mn, although the *T*_b_ has an opposite trend. Nevertheless, this observation points to the importance of correct statistical weighting of the magnetic parameters. In addition, the results corroborate the hypothesis by Ota and Takemura [51], who suggested the reduction of magnetization in the Néel regime (directly related to the μ_m_ and somewhat mirrored in the D_MAG_) due to effects reducing the effective magnetic volume of the MNPs. In our case, the D_MAG_ values are consistently lower for the Co1@Fe and Co1@Mn comparing to the Co2@Fe and Co2@Mn, and the “best heater”, Co2@Fe features both the largest μ_m_ and D_MAG_. In addition, this sample reveals heating curves for all frequencies without the non-linearities at the initial heating phase. Hence, the mean magnetic moment appeared to be the most relevant parameter for evaluating the heating performance as it implicitly reflects all the peculiar competitions of the relaxation mechanisms and possible collective effects, such as inter-particle interactions.

## 4. Conclusions

In summary, we have demonstrated that spinel ferrite core-shell nanoarchitectures are convenient systems for fine-tuning the magnetic parameters keeping the particle size variations moderate. Magnetic properties of the core-shell MNPs in our study, such as saturation magnetization, particle size, anisotropy constant, blocking temperature, and relaxation times, do not show a straightforward correlation to the SPA. More importantly, this study discloses that evaluating heating efficiency by considering SPA at the initial stage of heating does not represent the actual heating efficiency. In addition, the trends in the heat release at elevated temperatures on large time scales do not clearly correlate with the physical particle size and the basic magnetic parameters. Nevertheless, the mean magnetic moment fairly reflects all the phenomena involved. We observed that for every sample, there exists an interval of frequencies (and temperatures) where the SPA vs. *T* deviates from the expected linear trend, and the formal SPA values are higher than predicted from the trends predicted by the available theories; this observation unambiguously suggest the temperature-dependent competition of the different heating mechanisms. This particular finding is essential for all possible applications (magnetic fluid hyperthermia, MF-assisted chemical synthesis, and catalysis) where control of the temperature rise matters and may even become critical for the particular process. Our study, thus, points to a demand on the paradigm shift in standardizing the heating properties of MNPs so far based on a single number evaluated at the initial stage of the process.

## Figures and Tables

**Figure 1 nanomaterials-11-02848-f001:**
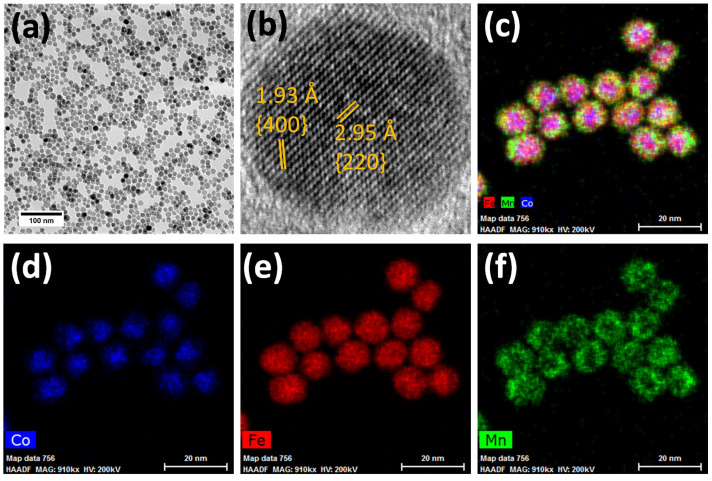
HR-TEM images and STEM-EDX maps across the oleate-capped Co1@Mn MNPs. Panel (**a**) shows a representative view of the sample, panel (**b**) demonstrates a well-crystalline structure of the MNP with lattice fringes and Miller’s indexes. A STEM-EDX joined map is shown in panel (**c**), cobalt, iron, and manganese STEM-EDX maps are presented in panels (**d**–**f**) respectively.

**Figure 2 nanomaterials-11-02848-f002:**
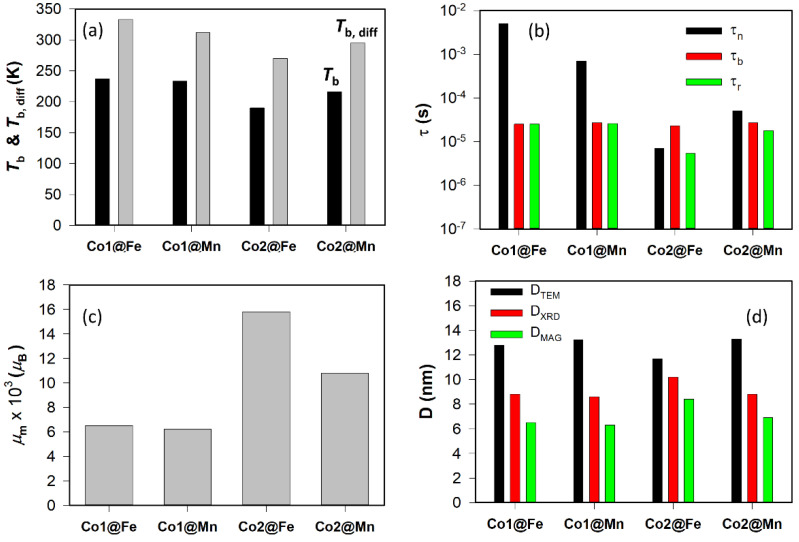
Comparison of the MNPs’ parameters for the Co1@Fe, Co1@Mn, Co2@Fe, and Co2@Mn samples. Mean blocking temperature (*T*_b_) and blocking temperature at the furcation of the ZFC and FC curves (*T*_b,diff_) (**a**), Néel (τ_n_), Brownian (τ_b_), and effective (τ_r_) relaxation times (**b**), mean magnetic moment (μ_m_) (**c**), and TEM (D_TEM_), XRD (D_XRD_) and magnetic (D_MAG_) diameters (**d**). The D_MAG_ values are based on the μ_m_ as a statistically relevant parameter (for details about the calculation, please, see Appendix A).

**Figure 3 nanomaterials-11-02848-f003:**
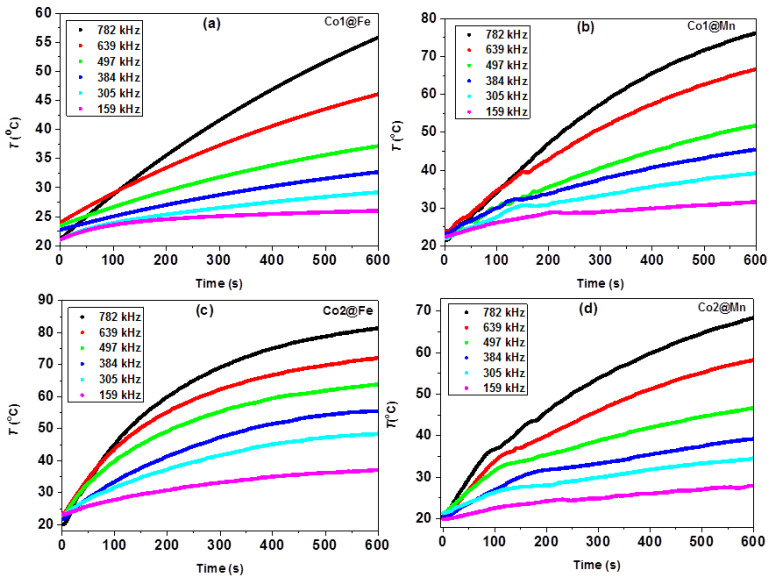
Temperature ramping of solution at different frequencies of AMF with field amplitude 31.6 mT (**a**–**d**).

**Figure 4 nanomaterials-11-02848-f004:**
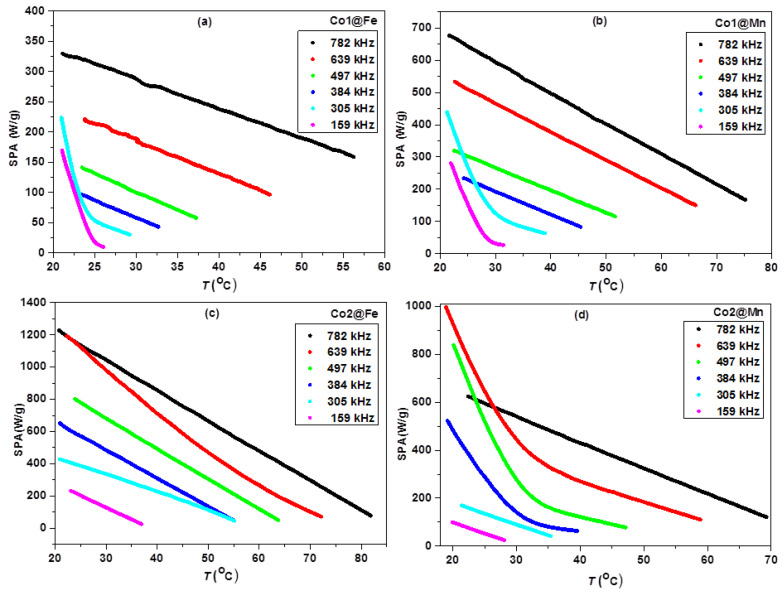
SPA vs. *T* at different frequencies for samples Co1@Fe (**a**), Co1@Mn (**b**), Co2@Fe (**c**) and Co2@Mn (**d**).

**Figure 5 nanomaterials-11-02848-f005:**
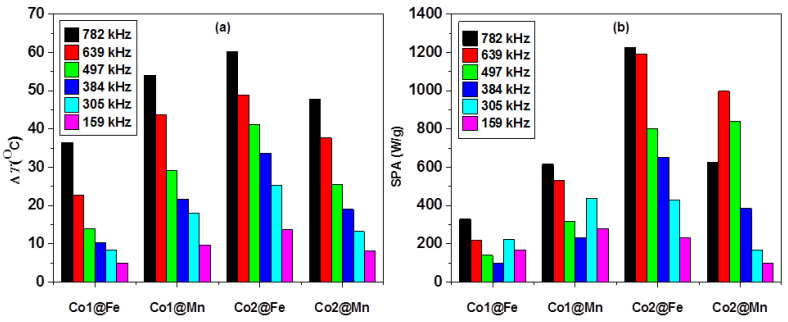
Variation of ∆*T* (**a**) and SPA (**b**) at frequencies indicated on legends and field amplitude of 31.6 mT.

## Data Availability

The data presented in this study are available on request from the corresponding author.

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
