# Peer review of "Self-Limitations of Heat Release in Coupled Core-Shell Spinel Ferrite Nanoparticles: Frequency, Time, and Temperature Dependencies"

_nanomaterials, 2021, doi:10.3390/nano11112848_

Round 1
Reviewer 1 Report
This paper reports the magnetic heating of core-shell spinels nano-particles.
I agree with their results and statements, however, there are some points to be revised.
- The introduction section seems to be too long. The authors should edit this section to make it shorter.
- How the core-shell structure improved thermal properties? I would like to know the thermal proerties of CoFe2O4 particles which do not have the shell and how improved.

Author Response
Dear Reviewer,
Thank you very much for your positive evaluation of our work. We are very grateful for your comments, which we carefully considered in our revision. All changes are depicted in the “tracked” version of the revised manuscript. A point-by-point reply to the comments is given below.
On behalf of authors, Jana Vejpravova
1. The introduction section seems to be too long. The authors should edit this section to make it shorter.
We agree that the Introduction can be reduced. In this vein, we did a significant reduction of the text and moved the technical parts to the SI.
2. How the core-shell structure improved thermal properties? I would like to know the thermal proerties of CoFe2O4 particles which do not have the shell and how improved.
Thank you very much for this comment. We extended the text in section 3.2:
"It is worth mentioning that the original cores (Co1 and Co2) show smaller ∆T (determined as a difference between the initial temperature and the final temperature after 600 s) comparing to the core-shell architectures. The effect is more pronounced for the Co2 series, where the ∆T recorded at the maximum frequency reaches about 10 K, while the core-shell MNPs reveal about 4 – 5 times larger values. The observed effect is usually attributed to the exchange coupling of the spins in the shell to the macrospin of the core giving rise to the enhanced heating performance [33]."
The data obtained for the Co1 and Co2 samples are given in the Supplementary Materials (Figures S3, S6, and S7).

Reviewer 2 Report
This manuscript was appropriately prepared and it can be accepted for publication in the current form. The subject herein discussed is of the interest for the reserach community working on the use of ferrite nanoparticles in hyperthermia studies.

Author Response
Dear Reviewer,
Thank you very much for your positive evaluation of our work. We believe that the revision based on the suggestions of the other two reviewers will still improve the original version.
On behalf of authors, Jana Vejpravova

Reviewer 3 Report
Dear authors,
after reading your manuscript i have come to the conlcusion that it needs some major revision, as indicated by the folliwng points:
- Introduction section is very wordy. Please consider shortening to the most relevant parts, e.g. by moving some subsections to the discussion part.
- I’d recommend to change eq. 5 to SPA =P/ρ= (current eq 5) /ρ. Then of course, respective text parts have to be changed accordingly
- Part 3.1: Better separation between previously published work and current new results.
- in section 3.1 you describe that oleate is used as capping agent, while CTAB was used in the methods part. Or were the images taken with samples that are not dispersible in water? In That case results may be misleading.
- Extrapolating the hydrodynamic size from literature for MNPs of comparable TEM size will most likely lead to wrong results. Please perform this simple DLS measurement to confirm your estimation.
- Please explain how you obtained D_TEM, D_XRD und D_Mag (n?, SD?)
- Heating curves: have you repeated the measurements? Are the shown values the mean of several measurement (n?)? If so what is the SD?
- Are the kinks in Fig. 3b and 3d really a particle based phenomenon or just an issue with data recording?
- How are the adiabatic conditions ensured and how did you avoid evaporation of the solvent at elevated temperature?
- What is the sample volume and particle concentration? What is kept constant between the different samples?
- I highly doubt your assumption that hydrodynamic size is “not expected to dramatically deviate among the series” and from personal experience, I know for sure that the hydrodynamic size changes during this relatively massive temp increase! To confirm your results, please perform a temperature dependent DLS measurement, then you can see if there’s a change or not.

Author Response
Dear Reviewer,
Thank you very much for your in-deep evaluation of our work. We are very grateful for your comments, which we carefully considered in our revision and which helped to improve the manuscript substantially. All changes are depicted in the “tracked” version of the revised manuscript. A point-by-point reply to the comments is given below.
On behalf of authors, Jana Vejpravova
- Introduction section is very wordy. Please consider shortening to the most relevant parts, e.g. by moving some subsections to the discussion part.
Thank you very much for this suggestion; another reviewer pointed out the same issue. Consequently, we did a significant reduction of the text and moved the technical parts to the SI.
- I’d recommend to change eq. 5 to SPA =P/ρ= (current eq 5) /ρ. Then of course, respective text parts have to be changed accordingly
We agree with this change, which will make the text clearer. We finally kept only the expression SPA =P/ρ in the main text.
- Part 3.1: Better separation between previously published work and current new results.
Thank you very much for this point. In the previous work, we focused mostly on the thorough structural and magnetic characterization of the MNPs by many complementary characterization techniques. The heating properties were measured in a homemade setup, which did not allow varying the magnetic field parameters (frequency, magnetic field). In the current work, we inspected the heating properties of the dispersions as a function of frequency. Also, we wanted to point out that the overall heating performance is not sufficiently characterized by the initial heating curve, due to the temperature change the heating mechanism may alter. We added a paragraph clearly stating this difference to the manuscript.
- in section 3.1 you describe that oleate is used as capping agent, while CTAB was used in the methods part. Or were the images taken with samples that are not dispersible in water? In That case results may be misleading.
The core-shell nanoparticles were synthesized through a two-step oleate-based solvothermal approach producing hydrophobic oleate-capped nanoparticles. The results shown in Figure 1 are referred to as the oleate-capped ones. Then, for the heat release tests, the particles were transferred into the water via intercalation of CTAB molecules. To make this part clearer, we modified the Experimental part and the caption of Figure 1. Moreover, thanks to the referee’s comment, we analyzed the sample Co1@Mn, reported in Figure 1, after the CTAB intercalation, to compare it with the oleate-capped one. We added the TEM images in the supporting information (Figure S1).
- Extrapolating the hydrodynamic size from literature for MNPs of comparable TEM size will most likely lead to wrong results. Please perform this simple DLS measurement to confirm your estimation.
Thank you very much for this comment. We have to admit that evaluation of the hydrodynamic size was the most critical point. The large set of the heating curves was obtained already prior to the lockdown of the laboratories during the pandemic. Currently, only the samples previously exposed to the magnetic field were available in the laboratories of the Charles University, where the heating experiments were performed. We made an attempt to measure the DLS, however, the results were not relevant for being included in the discussion as all samples sedimented and applying centrifugation and separation of the supernatant revealed no signal suggesting lack of individual MNPs in the dispersion. Keeping either the original concentration or very high dilution, and applying sonication to the dispersions, values in order of micrometers were obtained (please, see Additional supporting results – Figure AD1). Also, a fast test of heating abilities revealed much-reduced SPA values pointing to the irrelevance of using these data for the discussion in the manuscript.
Nevertheless, the team at the University of Cagliari performed DLS attempt on unexposed sample Co2@Fe using the original (aged) dispersion and that after separation of large aggregates (please, see the TEM images in Additional supporting results – Figure AD2 & Table AD1). As can be seen, the hydrodynamic size of the sample is in the range of 30-40 nm with a sigma of 8-10 nm. Thus, the Brown relaxation times in the range 7∙10-6-2∙10-5 regardless of the temperature (range 20-50°C). These values are in the range of those obtained for CTAB modified Zn-doped cobalt ferrite nanoparticles - 7.5 nm (TEM)/29 nm (DLS) comparing to the Co2@Fe - 11.7 nm(TEM)/32.8 nm(DLS). Still, considering the physical particle size for a series with the same concentration, identical coating (functionalization, amount of the organic content) one obtains a very good correlation between the DTEM and DH, which can be used as a very good estimate of the DH. A clear demonstration is shown in Additional supporting results – Figure AD3. Thus, this approach is still more appropriate comparing to using the DLS values recorded on the aggregated samples previously exposed to the magnetic field or samples with a different concentration of the MNPs.
In this vein, we modified the manuscript by adding relevant DH, tB, and teff values, recalculated all relevant dependencies and replotted the affected graphs (Figure 2(b), Figure S2). Nevertheless, the final picture is not changed by the revisions of the tB and teff values.
- Please explain how you obtained D_TEM, D_XRD und D_Mag (n?, SD?)
Thank you very much for this point. We included the technical details to the Experimental part, updated Table S1 in SI accordingly.
- Heating curves: have you repeated the measurements? Are the shown values the mean of several measurement (n?)? If so what is the SD?
In our experiments, we measured a frequency series of the heating curves starting from the highest available frequency and recording the next heating curve at a lower frequency after cooling down the sample. Then we reproduce the maximum and minimum frequency dependence and accept the results if the deviation does not exceed 10 %. We added all information to the manuscript (Experimental part).
- Are the kinks in Fig. 3b and 3d really a particle based phenomenon or just an issue with data recording?
Thank you very much for pointing this out. We were also quite a skeptical about the obtained data. Nevertheless, we were able to reproduce the curves very well. As an extreme example, we are including the case of the Co2@Mn sample, which we measured first in 2019 and later in 2021 when we started drafting the manuscript, please, see Figure AD4 in the Additional supporting results. Please, note that the second attempt was carried out on a sample not previously exposed to the magnetic field.
- How are the adiabatic conditions ensured and how did you avoid evaporation of the solvent at elevated temperature?
In our experiments, we are using a sealed vial and the temperature is recorded using an optical probe, which is immersed directly to the dispersion. The vial is placed in a glass jacket, which is continuously evacuated using a turbomolecular pump, which diminishes the heat convection to the surroundings. We added all information to the manuscript (Experimental part).
- What is the sample volume and particle concentration? What is kept constant between the different samples?
We are sorry for not giving this information directly. Yes, the concentration of the samples was fixed to 3.4 mg/ml. The volume used for the experiment was 2 ml (defined by the size of the standard vial). We added all information to the manuscript (Experimental part).
- I highly doubt your assumption that hydrodynamic size is “not expected to dramatically deviate among the series” and from personal experience, I know for sure that the hydrodynamic size changes during this relatively massive temp increase! To confirm your results, please perform a temperature dependent DLS measurement, then you can see if there’s a change or not.
Thank you very much for pointing this out. We verified the temperature effect on the sample that heat up the most (Co2@Fe) and we observed no significant variation of the hydrodynamic size in the temperature range 20-50 °C. The results were added to the SI (Figure S9 and Table S4). As can be seen, the changes are not dramatic within the experimental error, as expected giving rise to a rather moderate variation of the tb. (Also, the experiment on the aged and magnetic field exposed Co2@Fe sample revealed almost no change of the hydrodynamic size on temperature (Figure AD5).)

Round 2
Reviewer 3 Report
Dear authors, thank you for your effort to significantly improve the manuscript!